# Cadherin-11-Interleukin-6 Signaling between Cardiac Fibroblast and Cardiomyocyte Promotes Ventricular Remodeling in a Mouse Pressure Overload-Induced Heart Failure Model

**DOI:** 10.3390/ijms24076549

**Published:** 2023-03-31

**Authors:** Guojian Fang, Yingze Li, Jiali Yuan, Wei Cao, Shuai Song, Long Chen, Yuepeng Wang, Qunshan Wang

**Affiliations:** 1Department of Cardiology, Xinhua Hospital, School of Medicine, Shanghai Jiao Tong University, 1665 Kongjiang Road, Shanghai 200092, China; 2Department of Cardiovascular Surgery, Huadong Hospital Affiliated of Fudan University, 221 Yananxi Road, Shanghai 200040, China

**Keywords:** heart failure, Cadherin-11, IL-6, cardiac remodeling, inflammation

## Abstract

Heart failure is a serious and life-threatening disease worldwide. Cadherin-11 (Cad-11) is highly expressed in the heart and closely associated with inflammation. There is currently limited understanding on how Cad-11 contributes to cardiac remodeling and its underline molecular mechanism. We found an increased expression of Cad-11 in biopsy heart samples from heart failure patients, suggesting a link between Cad-11 and heart failure. To determine the role of Cad-11 in cardiac remodeling, Cad-11-deficient mice were used in a well-established mouse transverse aortic constriction (TAC) model. Loss of Cad11 greatly improved pressure overload-induced LV structural and electrical remodeling. IL (interleukin)-6 production was increased following TAC in WT mice and this increase was inhibited in cadherin-11^−/−^ mice. We further tested the effect of IL-6 on myocyte hypertrophy and fibrosis in a primary culture system. The addition of hCad-11-Fc to cultured cardiac fibroblasts increased IL-6 production and fibroblast cell activation, whereas neutralizing IL-6 with an IL-6 antibody resulted in alleviating the fibroblast activation induced by hCad-11-Fc. On the other hand, cardiomyocytes were promoted to cardiomyocyte hypertrophy when cultured in condition media collected from cardiac fibroblasts stimulated by hCad-11-Fc.Similarly, neutralizing IL-6 prevented cardiomyocyte hypertrophy. Finally, we found that MAPKs and CaMKII–STAT3 pathways were activated in both hCad-11-Fc stimulated fibroblasts and cardiomyocytes treated with hCad-11-Fc stimulated fibroblast condition medium. IL-6 neutralization inhibited such MAPK and CaMKII-STAT3 signaling activation. These data demonstrate that Cad-11 functions in pressure overload-induced ventricular remodeling through inducing IL-6 secretion from cardiac fibroblasts to modulate the pathophysiology of neighboring cardiomyocytes.

## 1. Introduction

Heart failure(HF) is a prevalent consequence of several cardiovascular diseases. It has become an increasingly serious public health problem accompanied by high hospitalization rates and mortality [1,2]. Cardiac remodeling refers to the change in cardiac structure and function, including perivascular and interstitial fibrosis, ventricular arrhythmias(VF) [3]. Cardiac remodeling can be classified into electrical remodeling and structural remodeling, based on the nature of the heart that is affected. Electrical remodeling refers to the change in electric properties such as ion channels that can lead to ventricular arrhythmias. Structural remodeling, on the other hand, refers to alteration in structure and composition of the heart that leads to cardiac fibrosis, ventricular hypertrophy, and heart failure [4,5]. Chronic inflammation has been linked to cardiac remodeling [6,7]. Recent studies have shown that multiple signal pathways play roles in both inflammation and cardiac remodeling [8]. Chronic inflammation is thus a prominent feature in cardiac remodeling, stimulating cytokine release.

Cadherin-11 (Cad-11), a member of the cadherin family, is a transmembrane protein mainly located at the cell junction that mediates homotypic cell-to-cell adhesion [9,10]. Research has shown that Cad-11 plays a critical role in inflammation-related diseases such as rheumatoid arthritis [11,12]. In cardiovascular disease, Cad-11 contributes to inflammation-driven cardiac fibrosis after MI by modulating both the recruitment of bone marrow-derived immune cells and their interactions with cardiac fibroblasts [13]. Our recent work demonstrated that Cad-11 participates in obesity-induced atrial fibrosis and susceptibility to atrial fibrillation by regulating the secretion of inflammatory factors IL-6 in atrial fibroblasts [14], implicating a role of Cad-11 in cardiovascular diseases.

Various cells including cardiac fibroblasts, can produce IL-6. IL-6 is a pro-inflammation cytokine that has been implicated in the pathogenesis of cardiovascular diseases [15,16]. Previous studies have suggested that IL-6 plays an important role in cardiac hypertrophy [17,18]. IL-6 can promote left ventricular hypertrophy without increasing blood pressure. Depletion of IL-6 can alleviate cardiac hypertrophy induced by Ang II and noradrenaline [15]. Thoracic aorta constriction (TAC)-induced hypertrophy and fibrosis in the LV are ameliorated by IL-6 ablation in mice [19], indicating that IL-6 plays a crucial role in cardiac hypertrophy caused by TAC.

In this study, we hypothesized that Cad-11 participates in TAC-induced cardiac remodeling by activating IL-6 signaling pathways. We use mouse TAC-induced cardiac remodeling to first investigate the effects of Cad-11 on pressure-overload-induced LV structural remodeling and electrical remodeling. We next studied the role of IL-6 in pressure-overload-induced LV hypertrophy, the direct effect of Cad-11 on IL-6 production in cardiac fibroblasts, and the potential role of Cad-11 in cardiac fibroblast (CF) activation. Finally, we explored the effect of Cad-11 on cardiac myocyte hypertrophy as well as its involvement in IL-6 signaling.

## 2. Results

### 2.1. Cad-11 Expression Is Up-Regulated in the Left Ventricles from Both DCM Patients and TAC-Induced Mice

To explore the correlation between Cad-11 and LV remodeling, we first examined Cad-11 expression levels in the left ventricles from dilated cardiomyopathy (DCM) patients by qRT-PCR and Western blotting. As shown in Figure 1, we observed elevated levels of Cad-11 both in messenger RNA and protein in the heart samples from DCM patients compared to those from normal donors (Figure 1A–C). The expression of atrial natriuretic peptide (ANP), brain natriuretic peptide (BNP) and β-myosin heavy chain (β-MHC) were also increased in LVs from DCM patients (Figure 1G). The transverse aortic constriction (TAC) is a commonly used mouse model for human DCM. We next determined whether Cad-11 expression is increased in the LV of TAC mice. Similar to DCM patient samples, LV tissues from WT mice four weeks after TAC had increased level of Cad-11 mRNA, as well as ANP, BNP, and β-MHC, compared to the sham-operated group (Figure 1D–F,H). These results suggest that Cad-11 might be involved in heart failure and LV remodeling.

### 2.2. Deletion of Cad-11 Protects against Pressure-Overload-Induced Cardiac Hypertrophy and Fibrosis in TAC Mice

To investigate the role of Cad-11 in the pathogenesis of cardiac remodeling, we used pressure-overload-induced LV structural remodeling model in WT and Cad-11-KO-mice. Heart tissues were harvested and analyzed from WT and Cad-11-KO mice 4 weeks post TAC. As shown in Figure 2A–D, in WT-TAC mice, a significantly enlarge in gross heart size as measured by the ratio of heart-weight/body-weight (2C) and heart-weight/tibia-length (2D) was observed when compared to sham groups (Figure 2A, top panel). H&E staining of heart cross-sections revealed the increase in cardiac wall thickness (Figure 2A, 2nd panel) in WT-TAC mice. WGA staining of heart cross-sections showed the enlarged cardiomyocytes in WT-TAC mice (Figure 2A, 3rd panel). In contrast, Cad-11^−/−^ mice showed only modest change in heart size, wall thickness, cardiomyocyte morphology after TAC treatment.

We further measured the mRNA levels of cardiac hypertrophy markers atrial natriuretic peptide (ANP), brain natriuretic peptide (BNP), and β-myosin heavy chain (β-MHC) in WT and Cad-11^−/−^ mice treated with TAC or sham. Consistent with above histological/morphological analysis, The increase in mRNA levels of all three markers in the heart from WT mice after TAC treatment were markedly reduced in Cad-11^−/−^-TAC mice (Figure 2E–G). These findings indicated that Cad-11 deletion reduced cardiac hypertrophy and fibrosis induced by pressure overload.

Histological analysis of heart cross-sections from WT-TAC mice stained with Masson’s trichrome revealed highly disorganized myofibers and increased collagen deposition (Figure 2H,I). Consistently, the profibrotic gene expression levels including collagen I, SMA, and CTGF were increased (Figure 2J,K). However, in Cad-11^−/−^-TAC mice, the heart cross-sections showed that the myofibers were less disorganized and collagen deposition were decreased, compare to that of WT-TAC mice (Figure 2H,I). The increased expression of profibrotic marker proteins in the heart of WT-TAC mice were attenuated when the Cad-11 gene was deleted (Figure 2J,K). These results indicated that Cad-11 participates in cardiac hypertrophy and fibrosis during TAC, suggesting its role in cardiac structural remodeling.

### 2.3. Deletion of Cad-11 Mitigates the Pressure-Overload-Induced Prolongation of Action Potential Duration (APD) and the Change in Corresponding Current Densities in Ventricle Myocytes

We next investigated the effects of Cad-11 deletion on pressure-overload-induced LV electrical remodeling. Action potential (AP) in the ventricular cardiomyocytes from the indicated mice was recorded using patch clamp technique (Figure 3A). To assess the health of the electrical activity of the heart, we measured the duration of the action potential (APD) in those mice. The APD20, APD50, and APD90 were dramatically prolonged in WT-TAC mice compared to WT-Sham mice (Figure 3B–D). The prolonged APDs were significantly attenuated in Cad-11^−/−^ TAC mice at APD50 and APD90. These results indicated that Cad-11 deficiency could markedly mitigate the prolongation of APD of the ventricular cardio-myocytes after TAC operation. In cardiomyocytes, prolonged APD results from either increased inward ionic flow or decreased outward current [20]. We therefore assess the effect of Cad-11 deletion on ion influx and outflux after TAC treatment.

Calcium influx (I_Ca++_) is known to be the main consequence upon activation of L-typed Calcium channel (LTCC)s, we therefore focused on LTCC. As shown in Figure 3E–G, the maximum current density in the WT-TAC mice was significantly lower than in the WT-Sham mice. The I_Ca-L_ amplitude was significantly reduced in the Cad-11^−/−^-TAC mice compared to in the WT-TAC mice. No significant differences were found between WT-Sham and Cad-11^−/−^-Sham mice (Figure 3E–G). This indicated that the abnormality in heart electrical activities caused by TAC was markedly relieved by Cad-11 deletion.

K+ current (I_K+_), on the other hand, has been shown to be closely related to repolarization in WT myocardium. The decrease in I_K+_ density can lead to acquired long QT syndrome in LVH and HF patients. We therefore recorded the K+ current in the cardiomyocyte isolated from the four indicated mouse groups to examine the change in I_k+_ and its three compoents I_tof_, I_Kslow_, and Iss. Figure 3H shows representative I_K+_ density record in LV cardiomyocytes. The mean I-V curves for I_K+_ I_tof_, I_Kslow_, and Iss densities were summarized in 3I. A significant reduction in I_K+_ densities was observed between WT-TAC and sham groups. By contrast, the reduction in I_K+_ densities was less in the Cad-11^−/−^-TAC group compared to that of WT-TAC group. Consistently, the mean values of I_tof_, I_Kslow_, and Iss densities decreased significantly in the WT-TAC group when compared to the WT-Sham and Cad-11^−/−^-Sham groups. The Cad-11 deletion alleviated the reduction by TAC operation, as demonstrated by one-way ANOVA of Itof, Ikslow, and Iss densities followed by Bonferroni’s multiple comparisons test at +50 mV.

Further, to determine whether Cad-11 deletion affects cardiac repolarization ion channel gene expression levels under TAC treatment, we employed qRT-PCR to measure the mRNA levels of ion channel genes Cav1.2 (for I_Ca++_), Kv4.2/Kv4.3/KCHIP2 (for Itof), and Kv1.5/Kv2.1 (for Ikslow) genes. Consistent with the above findings, the expression of all those ion channel genes were significantly reduced in WT-TAC mice, and the reductions were alleviated with Cad-11 deletion (Figure 3J–L). Taken together, our data demonstrated that the abnormality in heart electrical activities caused by TAC was markedly relieved by Cad-11 deletion.

### 2.4. Cad-11 Deletion Decreased IL-6 Production in Transverse Aortic Constriction (TAC) Mice

Inflammation is a common response in the heart in the pressure-overloaded left ventricle (LV) model and can lead to cardiac remodeling (ref.). Therefore, we next wanted to examine the effects of Cad-11 deletion on pressure-overload-induced LV inflammation. It has been shown that proinflammatory cytokines including MIF, TNF-a and IL-6 have been shown to be induced in response to the TAC mouse model. Similarly, we found a marked increase in mRNA levels of MIF, TNF-a, and IL-6 in WT-TAC mice (Figure 4A–C). The increase in those cytokines in WT-TAC mice were reduced in Cad-11^−/−^-TAC mice. The increase in IL-6 in WT-TAC mice was further validated by immunohistochemistry using IL-6 specific antibody staining on the heart cross-section samples from indicated mice (Figure 4D and summarized in Figure 4E), and IL-6 increase was inhibited by Cad-11 deletion in response to TAC. It has been known that IL-6 is released by cardiac fibroblasts in response to stimuli. We use immunofluorescent co-staining with antibodies against IL-6 and Vimentin, a marker for cardiac fibroblasts on heart cross-sections from WT or Cad-11^−/−^ mice with or without TAC respectively. As shown in Figure 4F, IL-6 expression was significantly higher in cardiac fibroblasts in WT-TAC mice than that in WT-Sham mice. Loss of Cad-11 significantly blocked IL-6 increase in cardiac fibroblasts in response to TAC. The attenuation of up-regulation of IL-6 in response to TAC in Cad-11-KO mice was further determined by Elisa assay (Figure 4G), showing an increase in IL-6 concentration in the serum of WT-TAC and such increase was attenuated in the serum of Cad-1^1−/−^ mice. Taken together, these data support a potential role of Cad-11 in IL-6 infiltration in the cardiac fibroblasts.

### 2.5. IL-6 Is Involved in Cad-11-Induced Cardiac Fibroblast Activation

Based on the above studies, we next examine whether Cad-11 directly modulates IL-6 production. We used cultured mouse cardiac fibroblasts for this study. HCad-11Fc, a fusion protein consisting Cad-11 extracellular domain linked to the antibody Fc region, has been shown to activate Cad-11 in cardiac fibroblasts and promote their proliferation and migration [21]. As shown in Figure 5A,B, addition of hCad-11-Fc to the cultured cardiac fibroblasts significantly increased IL-6 production. This increase was hCad-11-Fc concentration dependent, indicating a direct effect of Cad activation on IL-6 production. We confirmed that hCad-11-Fc promoted cardiac fibroblasts migration 24 h after hCad-11-Fc treatment in a wound healing assay (Figure 5C). This accelerated migration was suppressed by pretreating the cells with an anti-IL-6 antibody (hCad-11-Fc+anti-IL-6 in Figure 5C,D). Activation of cardiac fibroblasts is accompanied by an increase in fibrosis-related gene expression including Col1α1, αSMA, and CTGF (Figure 5E,F, IgG vs. hCad-11Fc). Pre-treatment with the IL-6 antibody reduced the increase in those fibrosis gene expression stimulated by Cad-11 activation. These data indicate that Cad-11 induced cardiac fibroblast activation is through IL-6.

### 2.6. Fibrotic Cad11-IL-6 Signaling Pathway Mediates NMVM Hypertrophy through Paracrine Effect

The above experiments established a Cad11-IL-6 signaling pathway that governs cardiac fibrosis. Our vivo data showed that loss of Cad-11 attenuates TAC-induced cardiac fibrosis and hypertrophy. To investigate how Cad-11^−/−^ exerts its anti-hypertrophy role, we first examined whether the expression of Cad-11 was upregulated in cardiomyocytes by hypertrophic stimulation. We used a cultured neonatal mouse ventricular myocytes (NMVM) as a model system to address this question. When NMVM cells were stimulated with Ang-II, a pro-hypertrophic factor, no Cad-11 expression was observed, consistent with previous report that Cad-11 was minimally expressed in cardiomyocytes [22]. However, when Ang-II was added to the cultured atria fibroblasts, we saw a significantly increased cadherin-11 expression [23]. Since we found that hCad-11-Fc significantly increased the production of inflammatory cytokine IL-6 in fibroblasts, and IL-6 plays an important role in cardiac hypertrophy, it is reasonable to hypothesize that the IL-6 secreted by fibroblasts upon hCad-11-Fc stimulation directly modulates cardiac hypertrophy through paracrine mechanism. To test this hypothesis, we collected conditional media from cultured cardiac fibroblasts treated with IgG or hCad-11-Fc respectively as indicated. NMVM myocytes were grown in those conditional media for 24 h before assay. As expected, the NMVMs grown in hCad-11-Fc treated cardiac fibroblast conditioned medium significantly induced cardiomyocyte hypertrophy, as demonstrated by marked increases in hypertrophy marker genes ANP, BNP, and β-MHC mRNA levels (Figure 6C–E), as well as cell surface area (Figure 6A,B). Pretreatment of NMVM cells with anti-IL-6 antibody, however, inhibited such increases in hypertropic marker gene expression and myocyte cell surface area induced by Cad-11 activated cardiac fibroblasts. Together, these results indicate that Cad-11 activates cardiac hypertrophy through paracrine IL-6 signaling between fibroblasts and cardiomyocytes.

### 2.7. Activation of MAPK and CaMKII-Dependent Activation of STAT3 Pathways Contributes to Cad-11-Induced Cardiac Fibroblast Activation

MAPK and Akt signaling pathways have been shown to be involved in gp130-mediated cardiomyocyte hypertrophy. We previously reported that two kinases in the MAPK signaling pathway including extracellular signal-regulated kinase ERK and c-Jun N-terminal kinase (JNK) were activated by Cad-11 [14]. Here, we validated our previous studies in cultured cardiac fibroblasts stimulated with hCad-11-Fc (Figure 7A). To investigate the role of IL-6 in Cad-11-mediated fibroblast activation, we added an IL-6 antibody to the culture media and found that IL-6 blockage suppressed hCad-11-Fc-stimulated MAPK activation, as measured by protein levels of p-ERK1/2, p-JNK, and p-P38 under indicated treatment condition. These data support a role of IL-6 in Cad-11-mediated MAPK activation (Figure 7A,C). ERK, JNK, and p38 of the MAPK signaling were involved in this activation.

A recent study suggested that IL-6 activated STAT3 in a CaMKII-dependent manner. Upon activation, STAT3 is phosphorylated and hence translocated to the nucleus. We reasoned that in cardiac fibroblasts, Cad-11 mediated CaMKII dependent STAT3 activation is through IL-6. The expression of CaMKII, p-CaMKII, and p-STAT3 were determined by Western blot analysis (Figure 7B and summarized in Figure 7D). We found that cardiac fibroblasts stimulated by hCad-11-Fc showed increased phosphor-CaMKII and STAT3, indicating STAT3 activation. This activation was attenuated by pre-treatment with anti-IL-6 antibodies in the culture (Figure 7B,D).

### 2.8. IL-6-MAPK and IL-6–CaMKII-STAT3 Pathways Participate in Cad-11-Induced Cardiomyocyte Hypertrophy through a Paracrine Effect

Activation of MAPK signaling pathway is associated with cardiac hypertrophy. To address the involvement of IL-6 and MAPK in Cad-11 induced cardiomyocyte hypertrophy, we used a cultured NMVM system with conditional media collected from cardiac fibroblasts under various treatment as indicated (Figure 7E,F). We found that the conditioned medium collected from hCad-11-Fc stimulated cardiac fibroblasts profoundly increases ERK and JNK phosphorylation in NMVMs. The pre-treatment of anti-IL-6 antibody could inhibit such augments (Figure 7E, summarized in Figure 7G). Several studies have shown that IL-6 activates the CaMKII-STAT3 signaling pathway in cardiomyocytes. We next tested whether Cad-11 regulated cardiac hypertrophy via the CaMKII-STAT3 pathway. As shown in Figure 7F,H, we found that cardiomyocytes treated with a conditioned medium collected from hCad-11-Fc stimulated fibroblasts significantly upregulated the p-CaMKII and p-STAT3 levels. Anti-IL-6 antibody significantly attenuated such activation. Our results indicated that Cad-11 promoted cardiomyocyte hypertrophy through a paracrine way involving IL-6 modulating MAPK and CaMKII-STAT3 signaling pathways.

To determine the effect of Cad-11 in cardiac hypertrophy via MAPK and CaMKII-STAT3 pathways in vivo, we performed TAC in Cad-11^−/−^ and WT mice and MAPK and CaMKII activation was measured by Western blotting using heart samples from those mice (Figure 8A–D). We found phosphorylation levels of ERK, JNK, CaMKII, and STAT3 were markedly increased in TAC mouse hearts, indicating the activation of MAPK and CaMKII-Stat3 pathways. Loss of Cad-11 suppressed MAPK, and CaMKII-STAT3 activation in vivo.

## 3. Discussion

Utilizing a loss-of-function mouse model, we investigated the role of Cad-11 in pressure overload-induced LV structural and electrical remodeling. We found that Cad-11 expression was reduced in heart tissues from both heart failure mice and DCM patients. Loss of Cad-11 inhibits persistent pressure overload-induced structural and electrical remodeling of the LV of the heart, resulting in improvement of heart function in HF. Cad-11 acted on cardiac fibroblasts to secrete IL-6 hence promoting fibroblast migration and fibrosis-related protein synthesis. The IL-6 secreted by fibroblasts can also act on neighboring cardiomyocytes through paracrine action. We further identified that MAPK and CaMKII-STAT3 activation are involved in mediating the effects of IL-6 on both cardiac fibroblasts and cardiomyocytes. We provided evidence obtained from both in vitro cell culture system and in vivo TAC mouse model that Cad-11 plays a role via IL-6 in both myocardial hypertrophy and fibrosis of the left ventricle induced by pressure overload.

Continuous pressure overload-induced LV remodeling will result in irreversible cardiac fibrosis, characterized by decreased LV compliance and diastolic dysfunction. During this process, fibroblasts will be activated and release high amounts of proinflammatory and profibrotic paracrine factors, which promote the proliferation of fibroblasts ensuing in cardiac fibrosis [24,25]. A soluble mediator, IL-6 plays multiple roles in inflammation, immune system, hematopoiesis. It can be produced by a variety of cells including immune cells and stromal cells. Il-6 is closely associated with many cardiovascular diseases. Here we showed its expression was elevated in myocardial infarction as well as heart failure. IL-6 is essential in the development of cardiac fibrosis as demonstrated in previous studies. However, the mechanisms that promote IL-6 production are unclear [26]. In this study, we found that Cad-11 null fibroblasts impaired IL-6 production in response to TAC. In cultured fibroblasts, hCad-11-Fc activates IL-6 production. Blocking IL-6 prevented hCad-11-Fc from activating fibroblasts. Thus, Cad-11 is a key upstream inducer of IL-6, which in turn causes cardiac fibrosis.

Cardiomyocytes and fibroblasts are closely related. Their communication is essential for maintaining cardiac function and promoting normal cardiac function. As a secretory cytokine, IL-6 can affect both fibroblasts and cardiomyocytes via autocrine and paracrine mechanisms [8,27,28].Through myocyte-to-fibroblast paracrine pathways, IL-6 produced by cardiomyocytes can promote cardiac fibrosis. Likewise, through fibroblast-to-myocyte paracrine pathways, IL-6 produced by cardiac fibroblasts can also result in cardiomyocyte hypertrophy. It has been reported that Cad-11 is minimally expressed in cardiomyocytes under normal conditions [28]. Its expression is not stimulated by hypertrophic stimuli in cardiomyocytes. Contrary to cardiomyocytes, adjacent fibroblasts exhibit a remarkable increase in Cad-11 expression levels. Despite the adverse consequences of cardiac hypertrophy caused by pressure overload, loss of Cad-11 significantly reduced the symptom. The conditioned medium derived from hCad-11-Fc stimulated fibroblasts activated cardiomyocytes and pre-treatment with IL-6 antibody blocked this conditioned medium-induced heart hypertrophy. These findings suggest that Cad-11 can participate in the process of cardiac hypertrophy and IL-6 mediates the crosstalk between cardiac fibroblasts and cardiomyocytes, leading to cardiac hypertrophy.

STAT1, STAT3, and MAPK signaling cascades have been reported to be activated by IL-6 [29]. Cardiac hypertrophy and fibrosis are linked to the abnormality of major STAT family isoforms and MAPK pathways. IL-6 activates CaMKII, which contributes to STAT3 activation [17]. In our previous study, we found and validated that ERK and JNK are activated by Cad-11 upregulation in atrial fibroblasts. In this report, we demonstrated the role of IL-6 in ERK/JNK activation in cardiac fibrosis and hypertrophy. IL-6 activates the MAPK pathway in hCad-11-Fc-induced cardiac fibrosis. In addition, we found that Cad-11 stimulates the production of IL-6, thereby activating CaMKII and subsequently STAT3. Taken together, our data suggested that MAPK and CaMKII-STAT3 signaling pathways are downstream of Cad-11-induced cardiac fibrosis via modulating IL-6 secretion.

We also found that IL-6 is important for the activation of the MAPK pathway in cardiomyocytes and that blocking IL-6 inhibited cardiac hypertrophy. We further identified that cardiomyocytes cultured in hCad-11-Fc stimulated conditioned medium increased both CaMKII and STAT3 phosphorylation levels; this effect was prevented by IL-6 antibody treatment. The inhibiting CaMKII and STAT3 phosphorylation lead to attenuation in cardiac hypertrophy. Therefore, Cad-11 activates fibroblasts to produce IL-6 which in turn stimulates cardiomyocytes through paracrine means; with subsequent activation of MPAK and CaMKII-STAT3 pathways in the cardiomyocytes that leads to cardiac hypertrophy.

In conclusion, Cad-11 deletion played a protective role in ameliorating pressure-overload-induced LV remodeling via a paracrine effect of IL-6 from cardiac fibroblast-to-cardiomyocyte. IL-6 thus functions as a down-stream signal for Cad-11 to activate MAPKs and CaMKII-STAT3 signaling pathways in the pathogenesis of myocardial fibrosis and cardiomyocyte hypertrophy (Figure 9).

## 4. Materials and Methods

### 4.1. Human Left Ventricular Tissues

The samples of human hearts from heart failure patients were obtained from the left ventricles of DCM patients undergoing heart transplants in Xinhua Hospital. Control samples were obtained from the left ventricles of normal heart donors who died due to accidents. Written informed consent was collected from each family of prospective heart donors. The study was carried out in accordance with the Declaration of Helsinki and approved by the ethics committee of Shanghai Xinhua Hospital affiliated with Shanghai Jiao Tong University School of Medicine.

### 4.2. Animal Model

All mice were in C57BL/6J background and used at 6–8 weeks old. The mice transverse aortic constriction (TAC) were performed according to the protocol described by Song et al. [30]. Briefly, mice were anesthetized with 2% isoflurane inhalation. A 7–0 silk thread was tied around a 27-gauge de-sharpened needle placed along the thoracic aorta following the trans-sternal thoracotomy. The needle was then removed and left stenosis in the aortic lumen. Sham-operated mice underwent the same procedure without aortic constriction. All animal experiments were approved by the Ethics Committee of Shanghai Xinhua Hospital affiliated to Shanghai Jiao Tong University School of Medicine.

### 4.3. Neonatal Mouse Ventricular Myocyte (NMVMs) and Neonatal Mouse Cardiac Fibroblast (CFs) Primary Culture

NMVMs or CFs were isolated and cultured from 1–3-day-old C57BL/6 mice, as previously described [31]. Briefly, heart tissues taken out from decapitated neonatal mice were washed in PBS and minced with scissor into small pieces followed by enzymatic disassociation in 0.25% trypsin-EDTA in PBS at 37 °C. Medium containing fetal bovine serum (FBS) was added to the isolated cells and pelleted by centrifugation at 1000 rpm for 5 min. The pelleted cells were re-suspended in a medium containing 10% fetal bovine serum and 1% penicillin/streptomycin and pre-plated for 30 min at 37 °C. Pre-plating facilitates cardiac fibroblasts to adhere to the culture dish, separating them from cardiomyocytes that floated in the suspension. CFs were attached to culture dishes within 2–3 h. The cardiomyocytes in the supernatant were collected, re-pelleted, and re-suspended in DMEM containing 10% FBS, 1% penicillin/streptomycin, and bromodeoxyuridine (1:100, to inhibit fibroblast growth). A total of ~2 × 10^6^ cells were hence seeded in a 65-mm culture dish. When the cell populations reached ~80% confluence, the cell cultures (CFs/NMVMs) were serum starved for 24 h followed by addition to hCad-11-Fc or conditional medium for the indicated times and concentrations. For inhibition related experiments, the indicated inhibitors [IL-6 neutralizing antibody (R&D, USA)] were added to the media accordingly.

### 4.4. Isolation of Adult Mouse Ventricular Myocytes (AMVMs)

Primary adult mice ventricular cardiomyocytes were isolated using protocol described by Shuai et al. [32]. In brief, mice were heparinized and sacrificed by cervical dislocation. The hearts were rapidly excised and connected to a modified Langendorff apparatus with a constant flow at a rate of 2–3 mL/min for retrograde perfusion through the aorta, for 3–5 min in calcium-free Tyrode’s solution (mM: NaCl 136; KCl 5.4; HEPES 10; NaH2PO4 0.33; MgCl2 1.0; and glucose 10; pH adjusted to7.4 with NaOH). The solution was switched to a digestive solution containing 0.3 mg/mL collagenase type II (Sigma, USA), 0.1% bovine serum albumin, and 30 μM of CaCl_2_ for 15–20 min. The left ventricular (LV) free wall was dissected from the hearts and placed in KB solution (mM: KOH 85; K-glutamate 50; taurine 20; KCl 30; MgCl_2_ 1.0; EGTA 0.5; HEPES 10; and glucose 10, with pH adjusted to 7.4 with KOH) at 4 °C until the patch-clamp recording was done. Only single rod-shaped cells with clear cross-striations were used for electrophysiological recording.

### 4.5. Cellular Electrophysiology Recording

Whole-cell patch-clamp was performed using the recording technique (Axopatch700B amplifier; Axon Instruments, Inc., Union City, CA, USA) as previously described [32]. Cells were firstly placed in a 1-mL chamber and superfused with an external solution at 2 mL/min before being clamped and another 5 min were required to stabilize the cells. Clampfit 10.7 and Origin 9.0. were used to analyze the recording data.

For action potential (AP) recordings, the isolated ventricular cardiomyocytes were continuously superfused with the normal Tyrode solution. Patch pipettes with tip resistance of 3 to 5 MΩ were filled with pipette solution containing (mmol/L): K-aspartate 110; KCl 30; NaCl 5; HEPES 10; EGTA 0.1; MgATP 5; creatine phosphate 5; and cAMP 0.05, pH 7.2 with KOH. APs were elicited by 2-millisecond-duration, 1000- to 2000-pA rectangular pulses at a basic cycle length of 1 s. Then the AP parameters were analyzed at PD20, APD50, and APD90, respectively.

For ICa-L recordings, the isolated ventricular cardiomyocytes were superfused with an external solution containing (mmol/L) C_5_H_14_ClNO 100; NaCl 35; NaH_2_PO_4_ 0.33; MgCl_2_ 1; KCl 5.4; CaCl_2_ 1.8; HEPES 10; glucose 10; BaCl_2_ 0.1; 4-aminopyridine 5, and Ph 7.4 with NaOH. Patch pipettes with tip resistance of 3 to 5 MΩ were filled with pipette solution containing (mmol/L) CsCl 120; EGTA 10; CaCl_2_ 1; MgCl_2_ 5; Na_2_-ATP 5; HEPES 10, and pH 7.2 with CsOH. ICa-L was elicited using a series of 200-millisecond steps from −40 to +50 mV with an increment of 10 mV from a holding potential of −80 mV before a 100-millisecond prepulse to −40 mV.

For IK recordings, the isolated cardiomyocytes were superfused with an external solution containing (mmol/L): NaCl 140; KCl 4; MgCl_2_ 1; NaH_2_PO_4_ 0.33, CaCl_2_ 1.8; HEPES 10; glucose 10, and pH adjusted to 7.4 with NaOH. The patch pipettes were filled with pipette solution containing (mmol/L) KCl 140; MgCl_2_ 1; HEPES 10; EGTA 10; Mg-ATP 5, and pH adjusted to 7.2 with KOH. Voltage-dependent potassium currents were elicited with voltage steps between −50 mV and +60 mV in 10-mv increments, for 4 s with a 10-s interval from a brief (20 ms) voltage step to −20 mV to activate and inactivate sodium channels. The decay phases of the outward potassium currents evoked during the 4.0 s depolarizing voltage step to +50 mV were fit by a double exponential function of the form: Y(t) = A1 exp(−t/τ1) + A2 exp(−t/τ2) +Ass where t is time, τ1 and τ2 are the decay time constants, A1 and A2 are the amplitudes of fast transient outward potassium currents (Ito, f), and slowly inactivates the potassium current (IK, slow), and Ass is the amplitude of the non-inactivating steady-state outward potassium current (Iss) [33].

### 4.6. Myocyte Surface Area Measurement

Well-spared NMVMs were grown in 6-well plates. The surface area of a single myocyte was measured at indicated time point using the Image Pro Plus Data Analysis Program [Media Cybernetics, USA]. The images of cardiomyocytes were captured by IX71 (Olympus, Japan), from randomly selected fields (50 for each group) of three independent experiments. The surface area of a fully extended individual cardiomyocyte within the visual field (under a 20× objective) was measured and the fold change was calculated by normalizing to the control group.

### 4.7. Cell Migration Assay

The migration of fibroblasts was measured by wound-healing assay. CFs were grown to confluence in 6-well plates and the bottom monolayer of cells was scraped away using a sterile p200 pipette tip. The remaining adhered cells were then treated with or without hCad-11-Fc and/or inhibitors in serum-free DMEM media for 24 h to allow cells migrating to the denuded (scraped) area. For each well, images of four to five randomly selected regions were captured at 0 and 24 h under an Olympus inverted microscope. The relative speed of CFs migration was calculated as the mean linear movement of fibroblasts over wound edges at 24 h. Further, migrated fibroblasts over time were normalized with the migration of untreated cells and expressed as a fold change from controls.

### 4.8. QRT-PCR

Total RNA was extracted using TRIzol reagent (Takara, UAS). cDNA was synthesized using qScript cDNA SuperMix (Takara, USA). qRT-PCR was performed using SYBR Green (Takara, USA) and normalized to glyceraldehyde-3-phosphate dehydrogenase (GAPDH) expression. The PCR primers were synthesized by Sangon Biotech and listed in Appendix A.

### 4.9. Western Blot

Heart tissues or cells (NMVMs and CFs) were homogenized and lysed in cold RIPA lysis buffer (Thermo Fisher Scientific, USA) containing a protease and Phosphatase inhibitor cocktail (Beyotime, USA). Total protein for each sample was quantified by BCA Protein Assay Kit. Proteins were separated by SDS-PAGE and transferred to PVDF membranes (Millipore, USA). Western blot were performed by blocking the membrane with 5% skimmed milk, followed overnight incubation at 4° with indicated primary antibodies against Cad-11 (Thermo Fisher Scientific), Collagen I (Abclonal), α-SMA (Abcam), CTGF (CST), p-ERK1/2 (CST), ERK1/2 (CST), p-JNK (CST), JNK (CST), p-P38 (CST), P38 (CST), p-CaMKII (CST), CaMKII (CST), p-STAT3 (CST), STAT3 (CST), GAPDH (Proteintech). After rigorous washing, the membranes were incubated with horseradish peroxidase-conjugated anti-rabbit/mouse secondary antibodies and visualized on a Gel Imaging System (Tanon, China). The band intensity was quantified by ImageJ software (1.51) after normalization to their respective GAPDH loading control.

### 4.10. Enzyme-Linked Immunosorbent Assay (ELISA)

The concentrations of IL-6 in the serum or medium were determined by ELISA kits (R&D Systems) according to the manufacturer’s protocol. HIgG was purchased from Sigma-Aldrich, USA.

### 4.11. Immunofluorescence

Heart tissues were fixed with 4% paraformaldehyde (PFA), embedded in tissue freezing medium (O.C.T.) blocks. 5 μm sections were cut for immunofluorescence assays. Primary antibodies against vimentin, IL-6, and ACTN2 were used for immunofluorescence. Sections were counterstained with 4,6-diamidino-2-phenylindole (DAPI) for nuclear stain. To measure cardiomyocyte cross-sectional areas, sections were incubated with FITC-labeled wheat germ agglutinin (WGA, L4895 from Sigma) and DAPI for 20 min, washed with PBS, then mounted. Confocal images were captured under a Zeiss LSM880 microscope with a 40× oil objective (Carl Zeiss, Germany). Cardiomyocyte cross-sectional areas were quantified using ImageJ software with at least 100 cells per animal. Four animals were studied per group.

### 4.12. Histological Analysis of Collagen Fibrosis

Heart tissues were fixed with 4% PFA, embedded in paraffin blocks. The paraffin sections (5 μm thickness) were stained with Masson trichrome solution to determine collagen deposition. At least 10 fields per animal were randomly selected in the ventricular tissue, and four animals were studied per group. The fraction of the blue stained area normalized to the total area was used as an indicator for myocardial fibrosis.

### 4.13. Statistical Analysis

The data are presented as mean ± SEM. Unpaired *t*-test, one-way analysis of variance test, and Turkey’s post-test were used to identify statistical significance. All data presented here were analyzed using SPSS 19.0 statistical software or GraphPad Prism 6.0. *p* < 0.05 was considered statistically significant.

## 5. Conclusions

Cadherin-11 play a role in pressure overload-induced ventricular remodeling via a cardiac fibroblast-to-cardiomyocyte paracrine effect. IL-6 was a downstream signal molecule of cadherin-11 and had a central role in myocardial fibrosis, and cardiomyocyte hypertrophy, which was mediated by activating the MAPKs and CaMKII-STAT3 pathways.

## Figures and Tables

**Figure 1 ijms-24-06549-f001:**
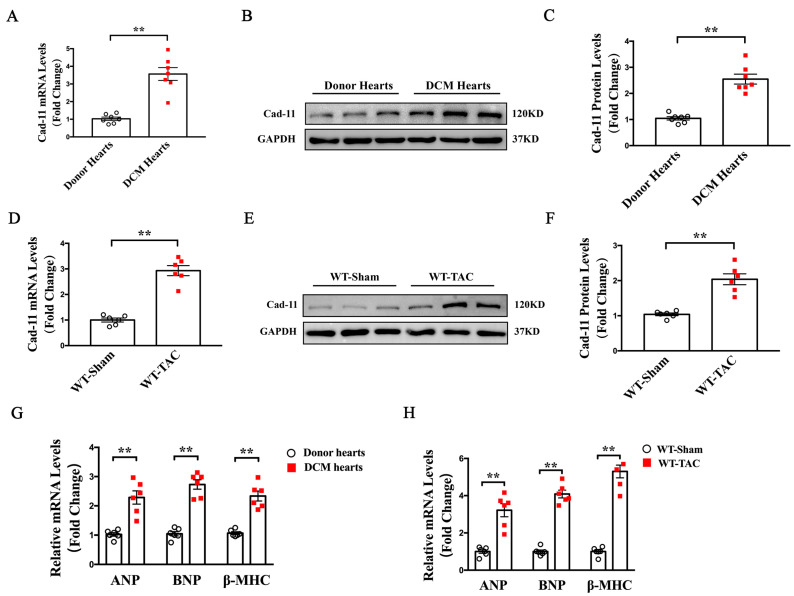
Increased Cad-11 expression was found in the left ventricles of heart from human heart failing patients and TAC-mice. (**A**–**C**) Representative mRNA and protein levels of Cad-11 in heart samples from normal donors and DCM patients. (**D**–**F**) Representative mRNA and protein levels of Cad-11 in the left ventricles of WT mice at 4 weeks after sham or TAC operation. (**G**) qRT-PCR analyses of ANP, BNP, and β-MHC mRNA levels in the left ventricles of normal donors and DCM patients (*n* = 6). (**H**) qRT-PCR analyses of ANP, BNP, and β-MHC mRNA levels in the left ventricles of WT-Sham and WT-TAC mice. Data were presented as mean ± SEM. ** *p* < 0.01 vs. Donor’s heart or WT-Sham group.

**Figure 2 ijms-24-06549-f002:**
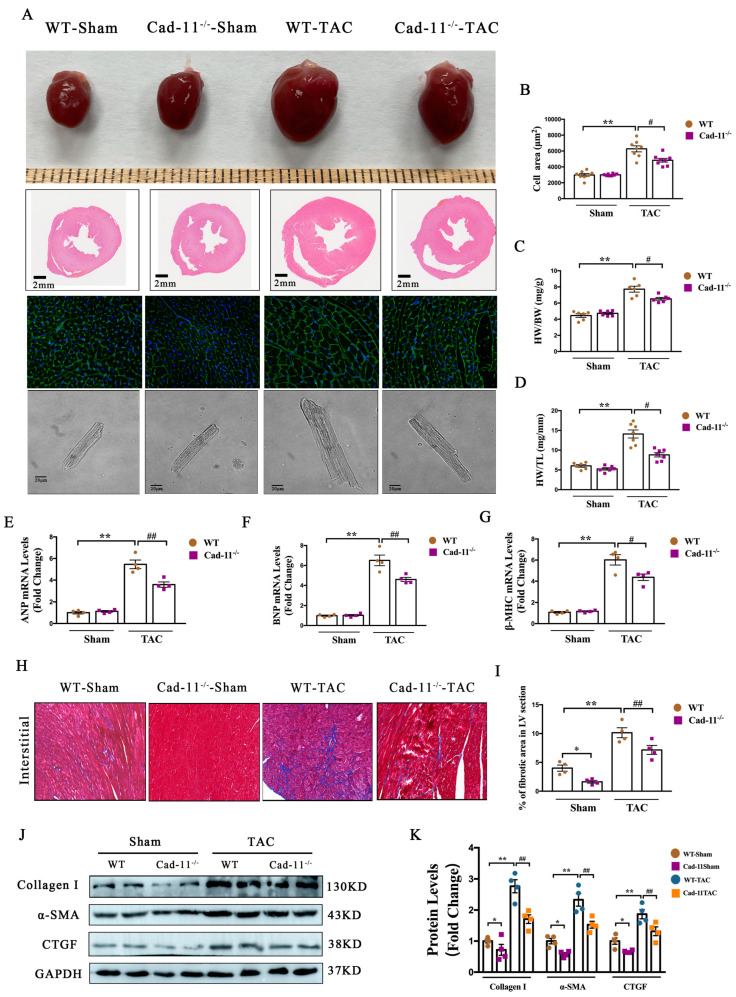
Loss of Cad-11 alleviated pressure overload-induced LV structural remodeling. (**A**) Representative images showing gross cardiac morphology used for heart weight/body weight (HW/BW) and heart weight/tibia length (HW/TL) calculation (top); transverse sections stained with H&E and WGA respectively (middle); Images of representative adult cardiomyocytes used for calculation of cell surface area (bottom). (**B**) Quantitative assessment of myocyte cell area calculated from isolated cardiac myocytes. (**C**,**D**) The ratio of heart weight/body vs. weight or heart weight/tibia vs. length were calculated from WT and Cad-11-knockout mice (KO) after sham or TAC operation. (**E**–**G**) Expression levels of cardiac hypertrophy–related genes ANP, BNP, and β-MHC were measured using quantitative reverse transcription polymerase chain reaction from heart samples of Sham- or TAC-mice 4 weeks after surgery. (**H**) Masson trichrome staining of representative heart sections from wild-type (WT) or Cad-11^−/−^ mice after sham or TAC operation. (**I**) Statistical analysis of differences in cardiac fibrosis. (*n* = 4 in each group). (**J**,**K**) Protein levels of fibrosis genes in the hearts from indicated mice determined by Western blotting. (*n* = 4 in each group). Data were presented as mean ± SEM, * *p* < 0.05, ** *p* < 0.01 vs. WT-Sham group. # *p* < 0.05, ## *p* < 0.01 vs. WT-TAC group.

**Figure 3 ijms-24-06549-f003:**
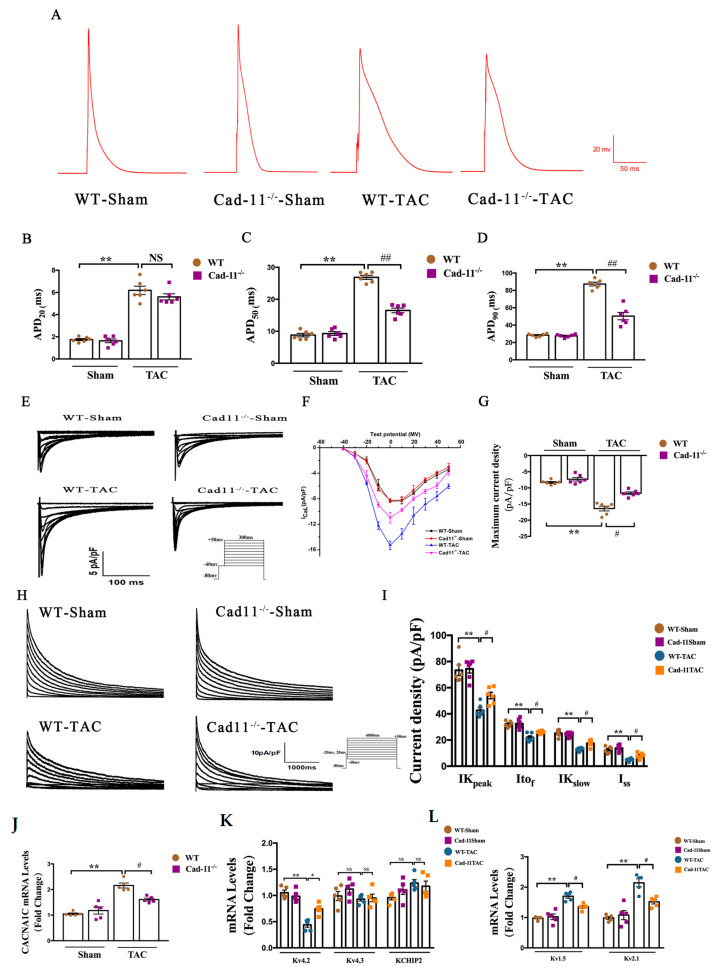
Cad-11 deficiency alleviated pressure overload-induced LV electrical remodeling. (**A**–**D**) Representative action potential curves for indicated mouse heart (**A**). (**B**–**D**) Statistical analysis of 20%, 50%, and 90% action potential durations (APDs) of LV myocytes isolated from Cad-11^−/−^ and WT mice that underwent sham or TAC surgery (*n* = 6 mice each group). (**E**) Representative chart of L-type calcium current tracings in indicated mice. (**F**) Representative I-V curve of Ica,L. Current density-voltage (I–V) relationship of L-type calcium current were measured using patch clamp. (**G**) Maximum L-type calcium current density in the LV myocytes of WT or Cad-11 KO mice 4 weeks after sham or TAC operation (*n* = 6 mice in each group). (**H**) Representative chart of voltage-dependent outward potassium current tracings. (**I**) Summary graph of the current density of outward potassium currents of WT and Cad-11 KO mice 4 weeks after sham or TAC operation (*n* = 6 mice in each group). (**J**–**L**) The mRNA levels of the calcium channel subunit Cav1.2, potassium channel subunits Kv4.2, Kv4.3 (for I_tof_), and Kv1.5, Kv2.1 (for I_Kslow_) determined by qRT-PCR. *N* = 4 mice in each group. Data were presented as mean ± SEM, ** *p* < 0.01 vs. WT-Sham group. # *p* < 0.05, ## *p* < 0.01 vs. WT-TAC group.

**Figure 4 ijms-24-06549-f004:**
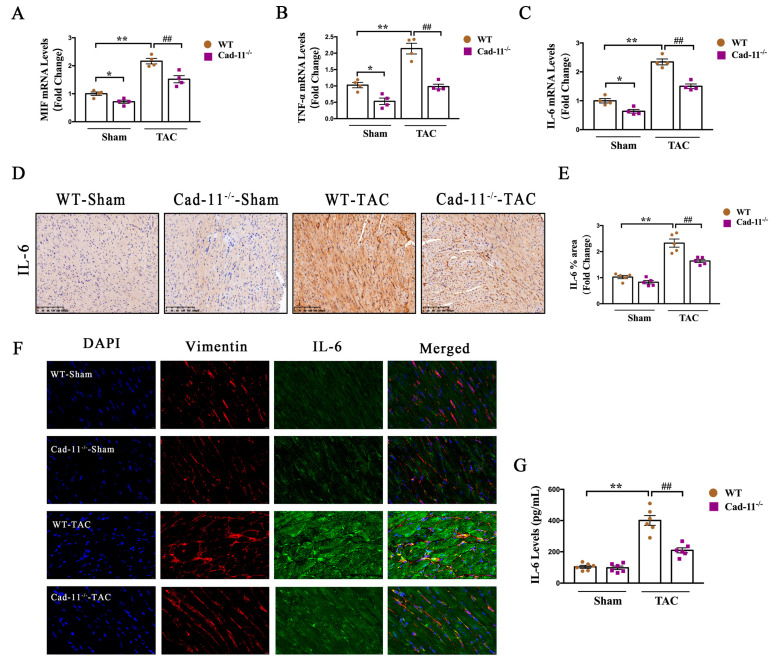
Loss of Cad-11 decreased IL-6 production in transverse aortic constriction (TAC) mice. (**A**–**C**) Relative mRNA levels of inflammation-related factors MIF, TNF-α, and IL-6 in samples obtained from WT and Cad-11^−/−^ mice LV tissues following 4 weeks of sham or TAC operation (*n* = 4 in each group). (**D**,**E**) Representative immunohistochemical staining of IL-6 using IL-6 antibody. Quantification of IL-6 expression in the LV of heart sections from the indicated mouse groups (*n* = 5 in each group) were summarized in (**E**). (**F**) Representative images of immunofluorescent staining of heart sections with anti-Vimentin (red), anti-IL-6 (green), and DAPI (blue) from WT-Sham, WT-TAC, Cad-11^−/−^-sham, and Cad-11^−/−^-TAC mice. (**G**) Average IL-6 concentrations (determined by ELISA) in serum in the indicated mouse groups. (*n* = 6 in each group). Data were presented as mean ± SEM, * *p* < 0.05, ** *p* < 0.01 vs. WT-Sham group. ## *p* < 0.01 vs. WT-TAC group.

**Figure 5 ijms-24-06549-f005:**
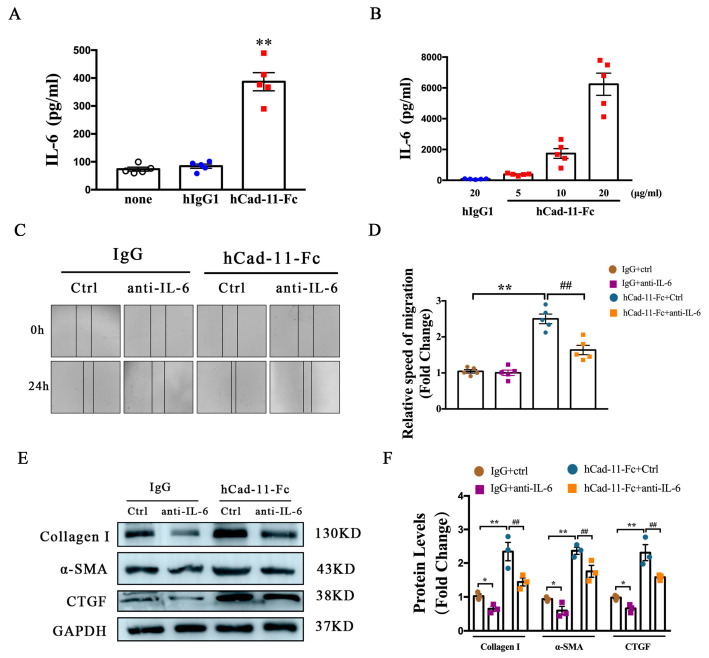
IL-6 participates in Cad-11-induced cardiac fibroblast activation. (**A**) The IL-6 protein levels determined by ELISA in primary cardiac fibroblasts treated with 5 μg/mL hCad-11-Fc, hIgG1, or PBS for 48 h. (**B**) Primary cardiac fibroblasts were cultured with 20 μg/mL hIgG1 or the indicated concentrations (μg/mL) of hCad-11-Fc for 48 h and the supernatants were analyzed for IL-6 by ELISA (*n* = 5 in each group). (**C**,**D**) Representative images of wound-healing assay in CFs captured at 0 and 24 h with or without the stimulation of hCad-11-Fc and anti-IL-6. Scale bar, 100 μm. The relative speed of fibroblast migration was measured by its mean linear movement over wound edges at 24 h (*n* = 5 in each group). (**E**,**F**) Representative immunoblots and statistics of fibrosis-related protein levels with or without the stimulation of hCad-11-Fc and anti-IL-6 measured by western blotting. Data were presented as mean ± SEM, * *p* < 0.05, ** *p* < 0.01 vs. the hIgG group, IgG+Ctrl group. ## *p* < 0.01 vs. hCad-11-Fc+Ctrl group.

**Figure 6 ijms-24-06549-f006:**
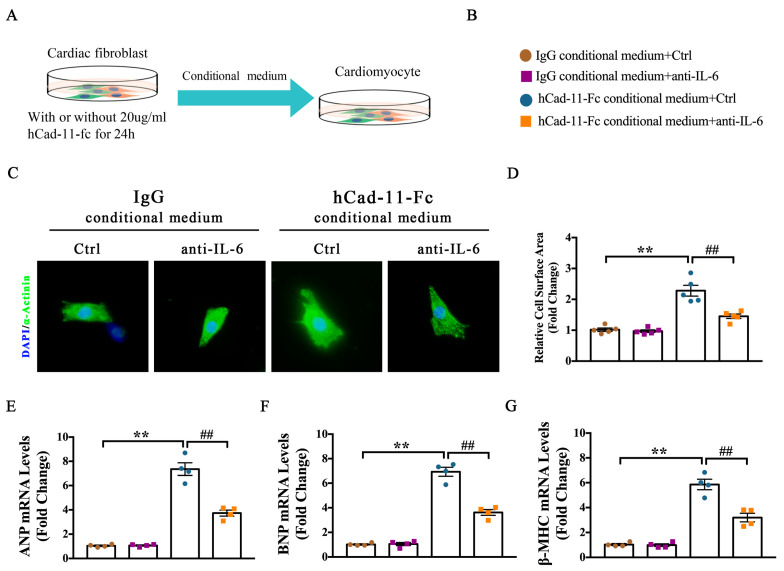
IL-6 mediates Cad-11-induced cardiomyocyte hypertrophy in NMVMs through a fibroblast-cardiomyocyte paracrine effect. (**A**) Schematic diagram of cardiac myocytes stimulated by conditioned medium collected from CFs infected with IgG or hCad-11-Fc. (**B**) Group information (**C**) Representative image of α-actinin staining for NMCMs from the indicated four mice groups. (**D**) Quantification of myocytes area for α-actinin staining of NMCMs in the indicated four mice groups (*n* = 5 in each group). (**E**–**G**) mRNA levels of cardiac hypertrophy–related genes ANP, BNP, and β-MHC using quantitative reverse transcription polymerase chain reaction from the indicated four groups of cells (*n* = 4 in each group). Data were presented as mean ± SEM, ** *p* < 0.01 vs. IgG conditional medium+Ctrl group. ## *p* < 0.01 vs. hCad-11-Fc conditional medium+Ctrl group.

**Figure 7 ijms-24-06549-f007:**
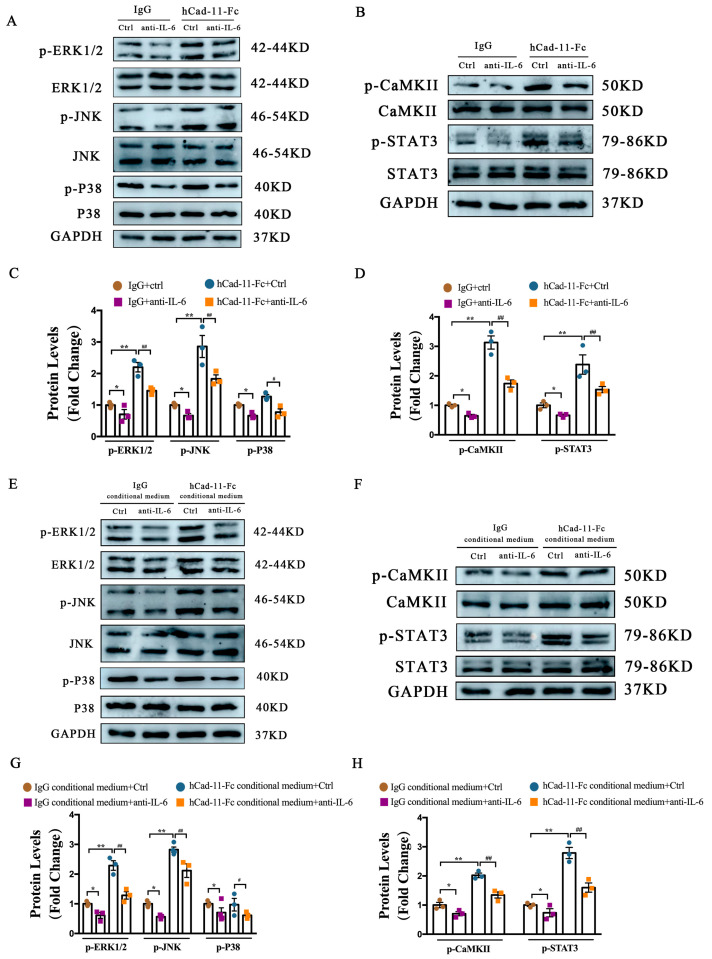
IL (interleukin)-6-activated MAPK and CaMKII–STAT3 pathways participate in Cad-11-induced cardiac fibroblasts activation and paracrine effect-induced cardiomyocyte hypertrophy. (**A**–**D**) Representative immunoblots and statistics showing phosphorylated ERK, JNK, P38 and phosphorylated CaMKII, STAT3 protein levels in neonatal mouse ventricular fibroblasts (CFs). The cells were pretreated for 30 min with or without anti-IL-6 antibody followed by incubation with IgG or hCad-11-Fc for 48 h (*n* = 3 in each group). (**E**–**H**) Representative immunoblots and statistics showing phosphorylated ERK, JNK, P38, and phosphorylated CaMKII, STAT3 protein levels in neonatal mouse ventricular myocytes (NMVMs) pretreated for 30 min with or without anti-IL-6 antibody followed by incubation with the conditioned medium collected from CFs stimulated with IgG or hCad-11-Fc. Data were presented as mean ± SEM, * *p* < 0.05, ** *p* < 0.01 vs. IgG+Ctrl group, IgG conditional medium+Ctrl group. # *p* < 0.05, ## *p* < 0.01 vs. hCad-11-Fc+Ctrl group, hCad-11-Fc conditional medium+Ctrl group.

**Figure 8 ijms-24-06549-f008:**
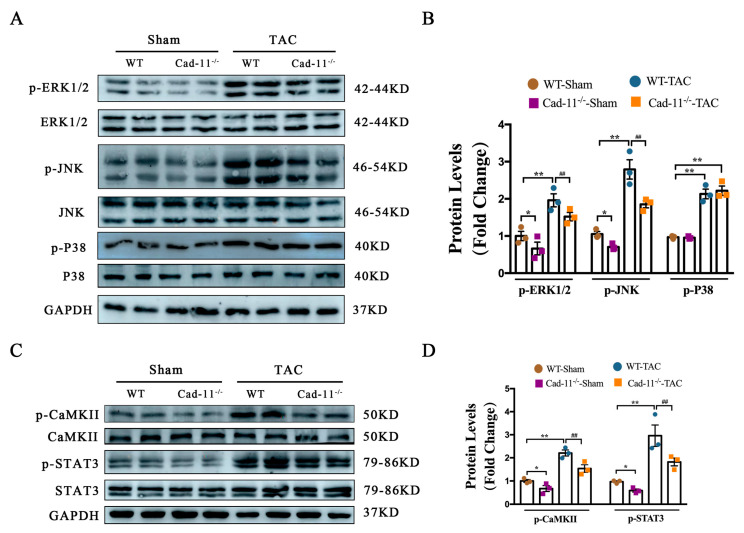
Loss of Cad-11 alleviates pressure overload-induced LV remodeling partially through MAPK and CaMKII–STAT3 pathways. (**A**,**B**) Representative western blotting and quantitative data of phosphorylation protein levels of ERK, JNK, and P38 (*n* = 3 in each group). (**C**,**D**) CaMKII, STAT3 in wild-type (WT) and Cad-11^−/−^ mice LV tissues following a 4-week sham or TAC operation (*n* = 3 in each group). Quantification of each protein expression level was normalized to GAPDH levels. Data were presented as mean ± SEM, * *p* < 0.05, ** *p* < 0.01 vs. WT-Sham group. ## *p* < 0.01 vs. WT-TAC group.

**Figure 9 ijms-24-06549-f009:**
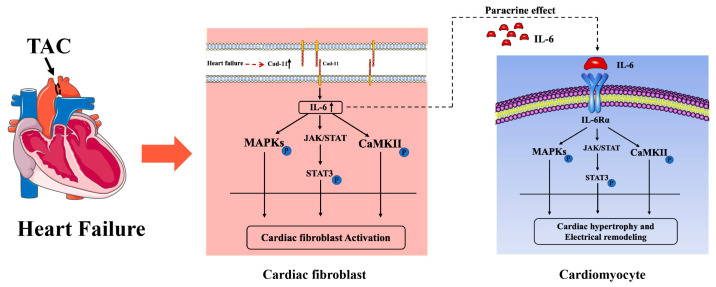
Model illustrating that Cad-11-IL-6 signaling pathway mediated cardiac fibrosis and myocardial hypertrophy in TAC-Induced Heart Failure.

## Data Availability

The datasets used and/or analyzed during the current study are available from the corresponding author on reasonable request.

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
