# Peer review of "Cadherin-11-Interleukin-6 Signaling between Cardiac Fibroblast and Cardiomyocyte Promotes Ventricular Remodeling in a Mouse Pressure Overload-Induced Heart Failure Model"

_ijms, 2023, doi:10.3390/ijms24076549_

Round 1
Reviewer 1 Report
Dear Authors,
First of all, I would like to congratulate you for the excellent technical-scientific work carried out by you, however, I found many typing errors such as, for example, punctuation, lack of space, etc., which must necessarily be corrected. In addition, I understand that the figures need to be revised and reformulated, as they are very small, with a lot of data and graphs that make it difficult to visualize and understand the results, so I ask you, please, to redo all the figures. so that in this way the results are valued, since they are very interesting.
Kind regards.
Author Response
Response to the reviewer1 comments:
Point 1: I found many typing errors such as, for example, punctuation, lack of space, etc., which must necessarily be corrected.
Response 1: Thank you for your valuable comments, which are very crucial in promoting our manuscript. We have carefully checked the language of the manuscript and correct errors such as punctuation and spaces. At the same time, an English-speaking colleague is also invited to make changes to the linguistic part of our manuscript, which has been marked with a red stamp in the manuscript.
Point2: In addition, I understand that the figures need to be revised and reformulated, as they are very small, with a lot of data and graphs that make it difficult to visualize and understand the results, so I ask you, please, to redo all the figures. so that in this way the results are valued, since they are very interesting.
Response 2: We are very grateful for your pertinent evaluation, and feel sorry for the pictorial problems you found in the manuscript. We have taken a closer check on the images submitted and found that this may have been caused by the improper layout of the images inserted into the word. Therefore, We've reworked and reformatted the images in the manuscript, enlarged the smaller graphs to ensure that the data and results are visible and identifiable, and uploaded all the images in another folder for you to view. If you have further questions, please do not hesitate to contact us in time, thank you!
Reviewer 2 Report
In the manuscript by Fang et al., the authors study the role of cadherin-11 in a murine model of cardiac hypertrophy. Cad-11 mice exhibit blunted structural and electrical remodeling in response to pressure overload-induced hypertrophy. The authors test the hypothesis that Cad-11 mediates maladaptive remodeling through fibroblast and cardiomyocyte crosstalk. Fibroblasts stimulated with recombinant Cad11 upregulate and secrete IL-6, which promotes myofibroblast differentiation in fibroblasts and hypertrophic signaling pathways in myocytes. The latter effect is mediated through activation of MAPK/CAMKII pathways. While this study sheds light on the downstream targets of Cad11 in the diseased heart, the discussion could use revisions for clarity.
Minor comments:
Lines 47 – 49: This a run-on sentence with many facts. Revise the grammar for clarity.
Line 58: First mention of “TAC” acronym but it is not defined until line 62. Define TAC here.
Line 313: “The role of Cadherin 11 in myocardial hypertrophy is mainly achieved by stimulating the secretion of IL-6 by fibroblasts…” This statement over interprets the data. The authors do show evidence of an IL6-MAPK/CAMKII-STAT3 pathway downstream of Cad11, but this is the only pathway tested. Cad11 may be functioning independent of IL6/MAPK in cardiac hypertrophy.
Revise lines 316-319 for clarity. Specifically:
“…activate the activity of fibroblasts” – be explicit on the “activity” in this statement
“And can further participate…” – what can participate?
“Eventually, participate in LV…” –Again, what is participating?
Line 321: “These fibroblasts become…” –which fibroblasts?
Line 326: “Many cardiovascular diseases produce IL-6”. Do the authors mean IL-6 is elevated in many cardiovascular diseases?
Line 328: Consider revising statement. This study found that Cad-11 null fibroblasts have impaired IL-6 production in response to TAC.
Line 334: Do the authors mean promoting normal cardiac function?
Line 337: It’s unclear what the authors mean by “myocyte-to-fibroblast effects”.
Line 371: Mentions “Figure 9” but there is no figure 9 in the main text.
Author Response
Response to the reviewer2 comments:
Point 1: Lines 47 – 49: This a run-on sentence with many facts. Revise the grammar for clarity.
Response 1: Thank you for your comments. Line 47-49, the statement "In cardiovascular disease, Cad-11 can regulate the recruitment of bone marrow derived cells, thereby limiting the expression of IL-6 induced by fibroblasts, promoting angiogenesis, thus participating in inflammation driven fibrosis remodeling after MI "was corrected as " Cad-11 contribute to inflammation-driven cardiac fibrosis after MI by regulating the recruitment of bone marrow-derived cells and their interactions with cardiac fibroblasts ", and you can see in Line 68-70.
Point 2: Line 58: First mention of “TAC” acronym but it is not defined until line 62. Define TAC here.
Response 2: We are sorry for this error and have been revised accordingly at line 81.
Point 3: Line 313: “The role of Cadherin 11 in myocardial hypertrophy is mainly achieved by stimulating the secretion of IL-6 by fibroblasts…” This statement over interprets the data. The authors do show evidence of an IL6-MAPK/CAMKII-STAT3 pathway downstream of Cad11, but this is the only pathway tested. Cad11 may be functioning independent of IL6/MAPK in cardiac hypertrophy.
Response 3: Thank you very much for your advice, we agree with your statement, and have modified the wording of this sentence based on your suggestion to change this sentence into “Cad-11 acted on cardiac fibroblasts to secrete IL-6 hence promoting fibroblast migration and fibrosis-related protein synthesis. The IL-6 secreted by fibroblasts can also act on neighboring cardiomyocytes through paracrine action. We further identified that MAPK and CaMKII-STAT3 activation are involved in mediating the effects of IL-6 on both cardiac fibroblasts and cardiomyocytes”. The exact location where the change can be found is in line 452-458 in the revised manuscript.
Point 4: Revise lines 316-319 for clarity. Specifically:
“…activate the activity of fibroblasts” – be explicit on the “activity” in this statement.
“And can further participate…” – what can participate?
“Eventually, participate in LV…” –Again, what is participating?
Response 4: We have re-written this part to “Cad-11 acted on cardiac fibroblasts to secrete IL-6 hence promoting fibroblast migration and fibrosis-related protein synthesis. Cad-11 induces IL-6 production in CFs, contributing to the activation of CFs and it was not stimulated in cardiomyocytes during cardiac hypertrophy. The IL-6 secreted by fibroblasts can also act on neighboring cardiomyocytes through paracrine action. We further identified that MAPK and CaMKII-STAT3 activation are involved in mediating the effects of IL-6 on both cardiac. In addition, fibroblasts and cardiomyocytes respond to IL-6 to a certain extent by activating MAPK and CaMKII-STAT3 signaling pathways. In general, Cad-11 can act on fibroblasts to secrete IL-6 and promote fibroblast migration and fibrosis-related protein synthesis through IL-6. Further, the secreted IL-6 can participate in myocardial hypertrophy through the paracrine effect. We provided evidence obtained from both in vitro cell culture system and in vivo TAC mouse model that Eventually, Cad-11 plays a role via IL-6 in both myocardial hypertrophy and fibrosis of the left ventricle induced by pressure overload" according to your valuable suggestion and also you can see the revised sentence in line 452-465.
Point 5: Line 321: “These fibroblasts become…” –which fibroblasts?
Response 5: We apologize for the misrepresentation. Sustained pressure overload-induced LV remodeling can promote myocardial fibrosis, a process in which fibroblasts are activated to accelerate myocardial fibrosis by releasing a variety of pro-inflammatory and fibrotic cytokines. Hence, these fibroblasts written in the text refer to fibroblasts that are activated during myocardial fibrosis. And we have revised the corrections in the manuscript, which you can see in line 466-470.
Point 6: Line 326: “Many cardiovascular diseases produce IL-6”. Do the authors mean IL-6 is elevated in many cardiovascular diseases?
Response 6: We are very sorry for the puzzle caused by the language problem. This is exactly what we want to say, and we have modified this sentence in line 472-474 of the text.
Point 7: Line 328: Consider revising statement. This study found that Cad-11 null fibroblasts have impaired IL-6 production in response to TAC.
Response 7: We sincerely appreciate your excellent suggestion. Line 328, the statement " This study found that Cad-11 ablation of TAC mice CFs inhibited IL-6 production largely" has been corrected as " This study found that Cad-11 null fibroblasts have impaired IL-6 production in response to TAC ", and you can see in line 476-477.
Point8: Line 334: Do the authors mean promoting normal cardiac function?
Response 8: We are sorry for this mistake. The correction has been made in line 482
Point 9: Line 337: It’s unclear what the authors mean by “myocyte-to-fibroblast effects”.
Response 9: Thank you for your comments. As we all know, IL-6, a secretory cytokine in the heart, can affect both fibroblasts and cardiomyocytes via autocrine and paracrine mechanisms. So, in myocyte-to-fibroblast paracrine pathways, IL-6 produced by cardiomyocytes can promote cardiac fibrosis. Likewise, in fibroblast-to-myocyte paracrine pathways, IL-6 produced by CFs can also result in cardiomyocyte hypertrophy. And you can see the context in lines 483-487 of the revision.
Point 10: Line 371: Mentions “Figure 9” but there is no figure 9 in the main text.
Response 10: We are sorry for this negligence. Figure 9 is the pattern diagram of the article, and it has been uploaded to the manuscript.
Round 2
Reviewer 1 Report
Dear Authors,
I congratulate you for the study work carried out and for the excellent article developed and, in addition, I am satisfied with the changes made to the article.
Kind regards,